artificial intelligence

peer-to-peer lending, artificial intelligence, big data, default risk, financial automation

**Author for correspondence:**
J. D. Turiel
e-mail: jeremy.turiel.18@ucl.ac.uk

# Peer-to-peer loan acceptance and default prediction with artificial intelligence

## J. D. Turiel[1] and T. Aste[1,2,3]

[1]Department of Computer Science, and [2]UCL Centre for Blockchain Technologies, University College London, Gower St, Bloomsbury, London WC1E 6BT, UK
[3]Systemic Risk Centre, London School of Economics and Political Sciences, Houghton Street, London WC2A 2AE, UK

JDT, 0000-0002-0451-6384; TA, 0000-0002-4219-0215

Logistic regression (LR) and support vector machine algorithms, together with linear and nonlinear deep neural networks (DNNs), are applied to lending data in order to replicate lender acceptance of loans and predict the likelihood of default of issued loans. A two-phase model is proposed; the first phase predicts loan rejection, while the second one predicts default risk for approved loans. LR was found to be the best performer for the first phase, with test set recall macro score of 77.4%. DNNs were applied to the second phase only, where they achieved best performance, with test set recall score of 72%, for defaults. This shows that artificial intelligence can improve current credit risk models reducing the default risk of issued loans by as much as 70%. The models were also applied to loans taken for small businesses alone. The first phase of the model performs significantly better when trained on the whole dataset. Instead, the second phase performs significantly better when trained on the small business subset. This suggests a potential discrepancy between how these loans are screened and how they should be analysed in terms of default prediction.

# 1. Introduction

Accurate prediction of default risk in lending has been a crucial theme for banks and other lenders for over a century. Modern-day availability of large datasets and open source data, together with advances in computational and algorithmic data analytics techniques, have renewed interest in this risk prediction task. Furthermore, automation of the loan approval processes opens new financing opportunities for small businesses and individuals. These previously suffered from limited access to credit, due to the high cost of human involvement in the process. Ultimately, automation of this process carries the potential to reduce human bias and corruption, making access to credit fairer for all. Financial technologies are having a strong impact on this

domain, which is rapidly changing [1]. The application of the model presented here to peer-to-peer (P2P) lending is just one example, indeed the present model can be directly applied to micro-financing in developing countries and loan-by-loan evaluation of loan investment portfolios.

P2P lending is defined as the practice of lending to individuals or businesses through an online platform that matches lenders with borrowers. These platforms often attempt to operate with lighter overhead than traditional financial institutions. This allows P2P lenders to provide higher returns to investors than traditional savings and investment accounts, even after fees are taken. With this type of lending, the default risk is often transferred to the investor and lifted from the lending platform. P2P lending has attracted the attention of industry, academics and the general public in recent years. This is also due to the large expansion of major P2P lending platforms like the Lending Club, which has now lent over $45 billion to more than 3 million customers. Another reason for the increasing coverage and popularity of P2P lending is its fast expansion to less developed markets in Eastern Europe, South America and Africa. As the monetary and social relevance of the industry grows, the need for regulation arises. The Financial Conduct Authority (FCA) is among the regulators which have set rules for this industry [2,3], indicating the importance of the trend in developed countries other than the USA.

Thanks to its easily accessible historical datasets, the Lending Club has been the subject of multiple publications investigating the drivers of default in P2P lending [4,5]. The growth of P2P lending in emerging countries has also attracted research interest, for instance [6] investigates lending in Mexico. This highlights the crucial role of P2P lending in providing access to credit for the population of emerging countries. Interdisciplinary scientific communities such as that of network science have started to show interest in the socio-economic dynamics of P2P lending [7]. More theoretical works have inquired about the reason for the need and growth of P2P lending. This was often connected to the concept of credit rationing due to asymmetric information between lending counterparts [8]. A solution to the problem of credit rationing, focused towards allowing fair access to credit and reducing poverty, are micro-finance institutions. Chris Anderson, Editor in Chief of *Wired* magazine, already identified in 2006 the concept of 'selling less of more', which is now making its way through to the lending market [9]. In order to reduce frictions and allow monetary financial institutions (MFIs) to have a self-sustainable business model, in [10] it was already suggested that technology will allow to reduce costs and interest rates, leading to an e-commerce-like revolution. This work aims to contribute to better understand the potentials and risks of automated lending technologies.

To the best of our knowledge, academic publications investigating the drivers of P2P lending [4–6] have applied simple regression models to this task. This work constitutes a significant step forward in applying big data and artificial intelligence techniques to P2P lending, combining two major disruptive emerging fields. The novelty and contribution of this work lies in the use of deep learning techniques, the introduction of an end-to-end model for loan issuance with the two phases described in §2 and the prediction-driven explainability of default drivers obtained from model analysis in §3.1.1.

The rest of the paper is organized as follows: in §2, we describe the dataset used for the analysis and the methods, in §3, we present results and related discussion for the first (§3.1.1) and second phase (§3.1.2) of the model applied to the entire dataset, §3.3 then investigates similar methods applied in the context of 'small business' loans, and §4 draws conclusion from our work.

# 2. Dataset and methods

## 2.1. Dataset

The data were collected from loans evaluated by Lending Club in the period between 2007 and 2017 (www.lendingclub.com). The dataset was downloaded from Kaggle (www.kaggle.com).

In this paper, we present the analysis of two rich open source datasets [11] reporting loans including credit card-related loans, weddings, house-related loans, loans taken on behalf of small businesses and others. One dataset contains loans that have been rejected by credit analysts, while the other, which includes a significantly higher number of features, represents loans that have been accepted and indicates their current status. Our analysis concerns both. The first dataset comprises over 16 million rejected loans, but has only nine features. The second dataset comprises over 1.6 million loans and it originally contained 150 features. We cleaned the datasets and combined them into a unique dataset containing ≈15 million loans, including ≈800 000 accepted loans. Almost 800 000 accepted loans labelled as 'current' were removed from the dataset, since no default or payment outcome was

available. The datasets were combined to obtain a dataset with loans which had been accepted and rejected and common features between the two datasets. This joint dataset allows to train the classifier for the first phase of the model: discerning between loans which analysts accept and loans which they reject. The dataset of accepted loans indicates the status of each loan. Loans which had a status of fully paid (over 600 000 loans) or defaulted (over 150 000 loans) were selected for the analysis and this feature was used as target label for default prediction. The fraction of issued to rejected loans is ≃10%, with the fraction of issued loans analysed constituting only ≈50% of the overall issued loans. This was due to the most recent loans being excluded, as well as those which have not yet defaulted or been fully paid. Defaulted loans represent 15–20% of the issued loans analysed.

In the present work, features for the first phase were reduced to those shared between the two datasets. For instance, geographical features (US state and postcode) for the loan applicant were excluded, even if they are likely to be informative. Features for the first phase are: (i) debt to income ratio (of the applicant), (ii) employment length (of the applicant), (iii) loan amount (of the loan currently requested), and (iv) purpose for which the loan is taken. In order to simulate realistic results for the test set, the data were sectioned according to the date associated with the loan. Most recent loans were used as test set, while earlier loans were used to train the model. This simulates the human process of learning by experience. In order to obtain a common feature for the date of both accepted and rejected loans, the issue date (for accepted loans) and the application date (for rejected loans) were assimilated into one date feature. This time-labelling approximation, which is allowed as time sections are only introduced to refine model testing, does not apply to the second phase of the model where all dates correspond to the issue date. All numeric features for both phases were scaled by removing the mean and scaling to unit variance. The scaler is trained on the training set alone and applied to both training and test sets, hence no information about the test set is contained in the scaler which could be leaked to the model.

Features considered for the second phase of the model are, (i) loan amount (of the loan currently requested), (ii) term (of the loan currently requested), (iii) instalment (of the loan currently requested), (iv) employment length (of the applicant), (v) home ownership (of the applicant; rented, owned or owned with a mortgage on the property), (vi) verification status of the income or income source (of the applicant; if this was verified by the Lending Club), (vii) purpose for which the loan is taken, (viii) debt to income ratio (of the applicant), (ix) earliest credit line in the record (of the applicant), (x) number of open credit lines (in applicant's credit file), (xi) number of derogatory public records (of the applicant), (xii) revolving line utilization rate (the amount of credit the borrower is using relative to all available revolving credit), (xiii) total number of credit lines (in applicant's credit file), (xiv) number of mortgage credit lines (in applicant's credit file), (xv) number of bankruptcies (in the applicant's public record), (xvi) logarithm of the applicant's annual income (the logarithm was taken for scaling purposes), (xvii) Fair Isaac Corporation (FICO) score (of the applicant), and (xviii) logarithm of total credit revolving balance (of the applicant).

We first analysed the dataset [11] feature by feature to check for distributions and relevant data imbalances. Features providing information for a restricted part of the dataset (less than 70%) were excluded and the missing data was filled by mean imputation. This should not relevantly affect our analysis as the cumulative mean imputation is below 10% of the overall feature data. Furthermore, statistics were calculated for samples of at least 10 000 loans each, so the imputation should not bias the results. A time-series representation of statistics on the dataset is shown in figure 1.

Differently from other analyses of this dataset (or of earlier versions of it, such as [12]), here for the analysis of defaults we use only features which are known to the lending institution prior to evaluating the loan and issuing it. For instance, some features which were found to be very relevant in other works [12] were excluded for this choice of field. Among the most relevant features not being considered here are interest rate and the grade assigned by the analysts of the Lending Club. Indeed, our study aims at finding features which would be relevant in default prediction and loan rejection *a priori*, for lending institutions. The scoring provided by a credit analyst as well as the interest rate offered by the Lending Club would not, hence, be relevant parameters in our analysis.

## 2.2. Methods

Two machine learning algorithms were applied to both datasets presented in §2.1: logistic regression (LR) with underlying linear kernel and support vector machines (SVMs) (see [13,14] for general references on these methodologies). Neural networks were also applied, but to default prediction only. Neural networks were applied in the form of a linear classifier (analogous, at least in principle, to LR) and a

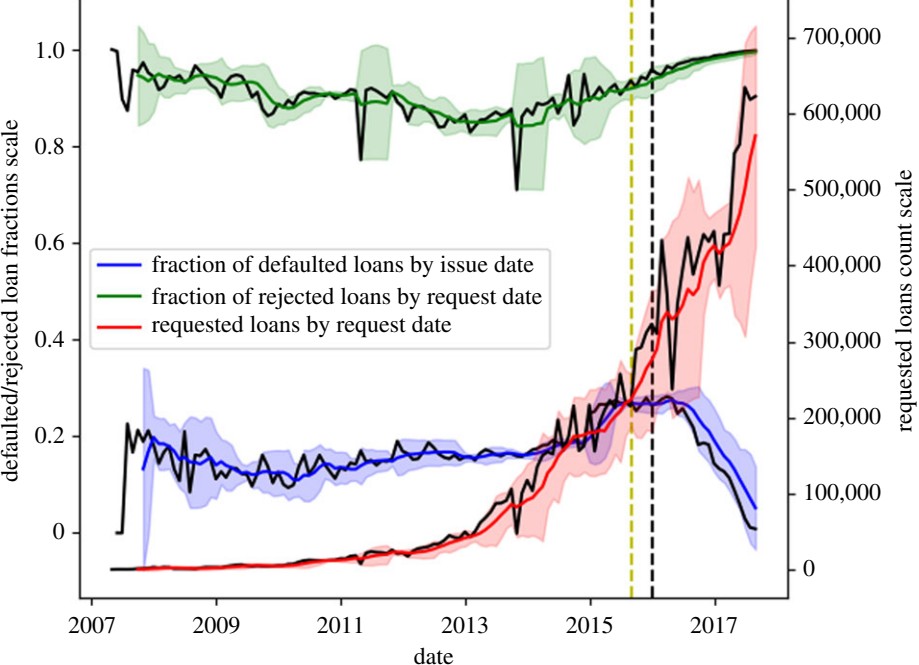

**Figure 1.** Time-series plots of the dataset [11]. Three plots are presented: the number of defaulted loans as a fraction of the total number of accepted loans (blue), the number of rejected loans as a fraction of the total number of loans requested (green) and the total number of requested loans (red). The black lines represent the raw time series, with statistics (fractions and total number) computed per calendar month. The coloured lines represent six-month moving averages and the shaded areas of the corresponding colours represent the standard deviation of the averaged data. The data on the right of the vertical black dotted line was excluded due to the clear decrease in the fraction of defaulted loans, this was argued to be due to the fact that defaults are a stochastic cumulative process and that, with loans of 36–60-month term, most loans issued in that period did not have the time to default yet. A larger fraction of loans is, instead, repaid early. This would have constituted a biased test set.

deep (two hidden layers) neural network [15]. A schematization of the two-phase model is presented in figure 2. This clarifies that models in the first phase are trained on the joint dataset of accepted and rejected loans to replicate the present decision of acceptance or rejectance. The accepted loans are then passed to models in the second phase, trained on accepted loans only, which improve on the first decision on the base of default probability.

### 2.2.1. First phase

Regularization techniques were applied to avoid overfitting in the LR and SVM models. L2 regularization was the most frequently applied, but also L1 regularization was included in the grid search over regularization parameters for LR and SVMs. These regularization techniques were considered as mutually exclusive options in the tuning, hence not in the form of an elastic net [16,17]. Initial hyperparameter tuning for these models was performed through extensive grid searches. The ranges for the regularization parameter $\alpha$ varied, but the widest range was $\alpha = [10^{-5}, 10^5]$. Values of $\alpha$ were of the form $\alpha = 10^n | n \in \mathbb{Z}$. Hyperparameters were mostly determined by the cross-validation grid search and were manually tuned only in some cases specified in §3. This was done by shifting the parameter range in the grid search or by setting a specific value for the hyperparameter. This was mostly done when there was evidence of overfitting from training and test set results from the grid search.

Class imbalance was mitigated through regularization as well as by balancing the weights at the time of training of the model itself.

Manual hyperparameter tuning was applied as a consequence of empirical evaluations of the model. Indeed, model evaluations through different measures often suggest that a higher or lower level of regularization may be optimal, this was then manually incorporated by fixing regularization parameters or reducing the grid search range. Intuition of the authors about the optimization task was also applied to prioritize maximization of a performance measure or balance between different performance measures. Due to data scarcity in this domain, training and test sets alone were used in

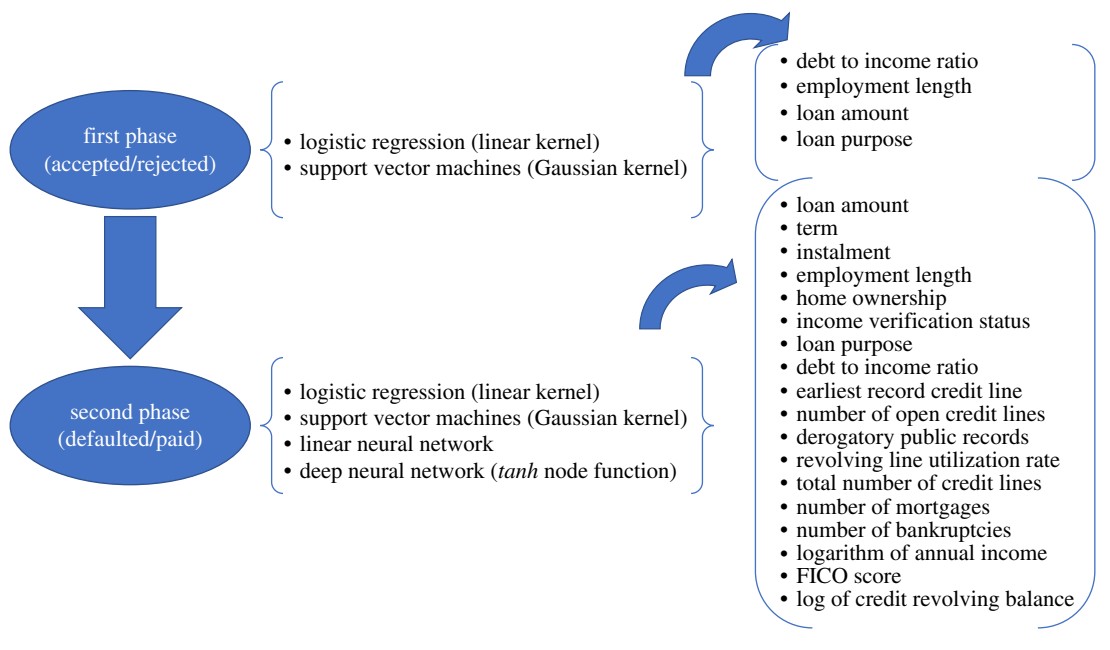

**Figure 2.** Scheme of the two-phase model for loan screening and default prediction (rating).

the analysis, with hyperparameter tuning performed through cross-validation. The dataset was split at the beginning in order to prevent information leakage, which might provide the model with information about the test set. The test set then contains future unseen data.

Two metrics were used for result validation, namely recall and area under the curve-receiver operating characteristic curve (AUC-ROC; see [18]). AUC-ROC can be interpreted as the probability that a classifier will rank a randomly chosen positive instance higher than a randomly chosen negative one [19]. This is very relevant to the analysis as credit risk and credit ranking are assessed in relation to other loans as well. The metric extrapolates whether defaulting loans are assigned a higher risk than fully paid loans, on average. Recall is the fraction of loans of a class (such as defaulted or fully paid loans) which are correctly classified. The standard threshold of 50% probability, for rounding up or down to one of the binary classes, was applied. This is relevant as it does not test the relative risk assigned to the loans, but the overall risk and the model's confidence in the prediction [20].

LR was applied to the combined datasets. The grid search over hyperparameter values was optimized to maximize the unweighted recall average. The unweighted recall average is referred to as recall macro and is calculated as the average of the recall scores of all classes in the target label. The average is not weighted by the number of counts corresponding to different classes in the target label. We maximize recall macro in the grid search as maximizing AUC-ROC led to overfitting the rejected class, which bares most of the weight in the dataset. This is due to AUC-ROC weighting accuracy as an average over predictions. This gives more weight to classes which are overrepresented in the training set, a bias that can lead to overfitting.

In order to obtain a more complete and representative test set, the split between training and test sets was 75%/25% for the first phase of the model (differently from the 90%/10% split applied in §3.1.2 for the second phase of the model). This provides 25% of the data for testing, corresponding to approximately two years of data. This indeed constitutes a more complete sample for testing and was observed to yield more stable and reliable results.

### 2.2.2. Second phase

Additional machine learning models were considered for this phase, namely linear and nonlinear neural networks with two hidden layers. Various choices had to be made in order to determine the activation function, optimizer, network structure, loss function and regularization method. We now outline the literature-based choices made and then move on to empirical hyperparameter tuning.

A *tanh* activation function was selected due to its widespread use in the literature for binary classification tasks. The choice was mainly between the *tanh* and *sigmoid* function, but as the former goes through zero with a steeper derivative, its backpropagation is usually more effective [21]. This was true in our case too.

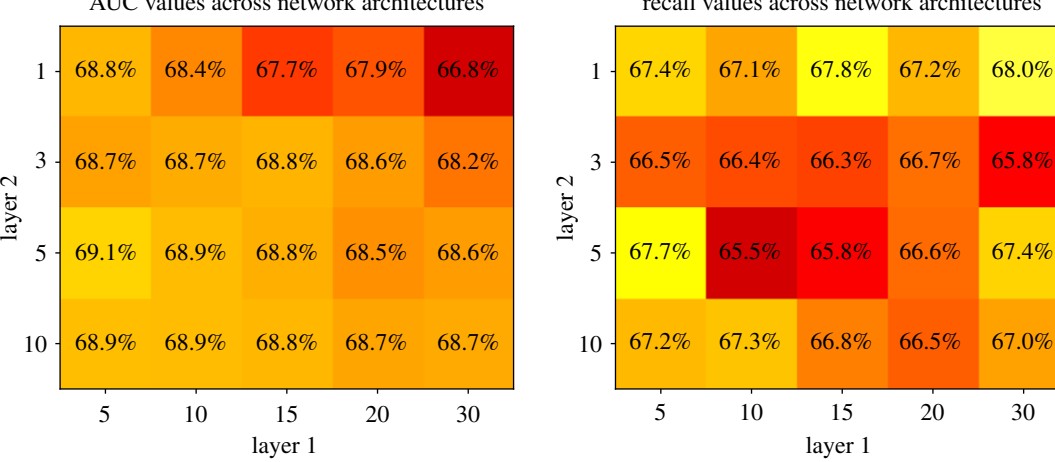

**Figure 3.** Stratified fivefold cross-validation grid search over network structures. The plots above represent labelled heatmaps of the average cross-validation AUC-ROC and recall values for the models. These were used to select the best performing architectures for which results are presented in table 2.

For optimization, the adaptive moment estimation (Adam) [22] optimization method was chosen. This was growing in popularity at the time of writing and it was designed specifically for neural networks. It should be noticed that Adam is a good paradigm for the class of adaptive gradient methods. Adam was shown to yield improvements in speed of training and performance as well as reducing the importance of learning rate tuning. Adam leverages adaptive learning to find learning rates tailored to each parameter. It incorporates benefits of adaptive gradient algorithm (AdaGrad) [23] and RMSprop [24]. Other methods were also tested and it was observed that regular stochastic gradient descent (SGD) methods with non-adaptive gradients presented worse out-of-sample performance.

The loss function used was 'softmax cross entropy' due to its widespread use in the literature, its interpretation in terms of probability distributions and the fact that it is agnostic to different activation functions and network structures [25].

Dropout was selected as a regularization method, as different features in lending data can often be missing or unreliable. Dropout regularizes the model while making it solid to missing or unreliable individual features. Consequences of this are discussed later in §3.2.

The network structure (number of nodes per layer) was then tuned through an empirical grid search over multiple network configurations, evaluated through stratified fivefold cross-validation [26] in order to avoid shrinking the training or test sets. A visualization of the mean AUC-ROC and recall values across folds for each configuration is shown in figure 3. The best models from these grid searches (DNN with [$n_1 = 5$, $n_2 = 5$] and DNN with [$n_1 = 30$, $n_2 = 1$]) are represented and matched with out-of-sample results in table 2.

LR, SVM and neural networks were applied to the dataset of accepted loans in order to predict defaults. This is, at least in principle, a much more complex prediction task as more features are involved and the intrinsic nature of the event (default or not) is both probabilistic and stochastic.

Categorical features are also present in this analysis. These were 'hot encoded' for the first two models, but were excluded from the neural network in this work as the number of columns resulting from the encoding greatly increased training time for the model. We shall investigate neural network models with these categorical features included, in future works.

For the second phase, the periods highlighted in figure 1 were used to split the dataset into training and test sets (with the last period excluded as per the figure caption). The split for the second phase was of 90%/10%, as more data improves stability of complex models. Balanced classes for model training had to be obtained through downsampling for the training set (downsampling was applied as oversampling was observed to cause the model to overfit the repeated data points).

In this phase, the overrepresented class in the dataset (fully paid loans) benefitted from the higher quantity of training data, at least in terms of recall score. In this case, the overrepresented class is that of fully paid loans while, as discussed in §3.1.1, we are more concerned with predicting defaulting loans well rather than with misclassifying a fully paid loan.

**Table 1.** Table with main results from LR and SVM tested for the second phase of the model.

| loan default prediction results | | | | |
| --- | --- | --- | --- | --- |
| model | $\alpha$ | recall train | AUC test | recall test (macro/default/paid) |
| LR | $10^{-2}$ | 64.3% | 69% | 63.7%/63.8%/63.6% |
| SVM | $10^{-2}$ | — | 64.3% | 62.2%/58.7%/65.6% |

# 3. Results and discussion

## 3.1. General two phases model for all purpose classes prediction

### 3.1.1. First phase

The grid search returned an optimal model with $\alpha \simeq 10^{-3}$. The recall macro score for the training set was $\simeq 79.8\%$. Test set predictions instead returned a recall macro score $\simeq 77.4\%$ and an AUC-ROC score $\simeq 86.5\%$. Test recall scores were $\simeq 85.7\%$ for rejected loans and $\simeq 69.1\%$ for accepted loans.

The same dataset and target label were analysed with SVMs. Analogously to the grid search for LR, recall macro was maximized. A grid search was applied to tune $\alpha$. Training recall macro was $\simeq 77.5\%$ while test recall macro was $\simeq 75.2\%$. Individual test recall scores were $\simeq 84.0\%$ for rejected loans and $\simeq 66.5\%$ for accepted ones. Test scores did not vary much, for the feasible range of $\alpha = [10^{-5}, 10^{-3}]$.

In both regressions, recall scores for accepted loans are lower by $\approx 15\%$, this is probably due to class imbalance (there is more data for rejected loans). This suggests that more training data would improve this score. From the above results, we observe that a class imbalance of almost $20\times$ affects the model's performance on the underrepresented class. This phenomenon is not particularly worrying in our analysis though, as the cost of lending to an unworthy borrower is much higher than that of not lending to a worthy one. Still, about 70% of borrowers classified by the Lending Club as worthy, obtain their loans.

The results for SVMs suggest that polynomial feature engineering would not improve results in this particular analysis. The surprisingly accurate results for LR suggest that credit analysts might be evaluating the data in the features with a linear-like function. This would explain the improvements shown by the second phase, when just a simple model was used for credit screening.

### 3.1.2. Second phase

LR, SVMs and neural networks were applied to the dataset of accepted loans in order to predict defaults.

#### 3.1.2.1. Second phase: logistic regression

The grid search for LR returned an optimal model with a value of $\alpha \simeq 10^{-2}$. The grid was set to maximize recall macro, as for the models in §3.1.1. Training recall macro score was $\simeq 64.3\%$ and test AUC-ROC and recall macro scores were 69.0% and 63.7%, respectively. Individual test recall scores were 63.8% for defaults and 63.6% for fully paid loans (table 1). Maximizing recall macro indeed yields surprisingly balanced recall scores for the two classes. Maximizing AUC-ROC did not lead to strong overfitting, differently from what is discussed in §3.1.1. Test scores were lower, both in terms of AUC-ROC and recall macro.

#### 3.1.2.2. Second phase: support vector machine

SVMs were also applied to the dataset. The optimal value of $\alpha$ returned by the grid search was $\alpha = 10^{-2}$, the same as for LR in §3.1.2—LR. Scores for the model were, though, worse than those returned by LR. Test AUC was $\simeq 64.3\%$ and individual test recall scores were 58.7% for defaulted loans and 65.6% for fully paid loans, see table 1. It can be inferred that the analysis of this dataset does not benefit from SVM kernel's nonlinearities in its test set performance. Furthermore, recall scores are improved for the overrepresented class in the dataset. This is the opposite of what is aimed for in this analysis, where we prioritize high recall on the default class which has a higher impact on the borrower's balance sheet. Such a strong score imbalance is also not ideal in terms of quality of the predictor. It should be

**Table 2.** Table with main results from DNN architectures tested for the second phase of the model.

| loan default prediction results | | | | |
| --- | --- | --- | --- | --- |
| model | dropout | recall train | AUC test | recall default test |
| DNN[a] | 20% | — | 68% | 67% |
| DNN[b] | 20% | 71% | 66% | 75% |
| DNN[c] | 20% | 68% | 69% | 72% |

[a]DNN with arbitrary node numbers [$n_1 = 20$, $n_2 = 5$].
[b]DNN with node numbers fine-tuned to [$n_1 = 30$, $n_2 = 1$].
[c]DNN with node numbers fine-tuned to [$n_1 = 5$, $n_2 = 5$].

noted that the label class imbalance (defaulted and fully paid loans) is much weaker than that described in §3.1.1, with defaulted loans representing 15–20% of the dataset.

### 3.1.2.3. Second phase: neural network

Linear neural network classifiers as well as deep (two hidden layers) neural networks were also trained on the dataset for the second phase of the model. Linear neural network classifiers were trained on numerical features alone as well as on both numerical and categorical features. L2 regularization was then applied. Numerical features-only test scores returned an AUC-ROC of 67.8% and a recall of 60.0% (for defaulted loans). The model yielded improved results when trained on categorical features too. Test scores returned an AUC-ROC of 68.7% and recall of 62.7% (for defaulted loans). These scores are slightly worse than those for LR, but they do not implement regularization yet. Once L2 regularization ($\alpha = 10$—this is a reasonable value commonly used in practice) was manually set and applied, test AUC-ROC improved to 69% and recall improved to 65% (for defaulted loans).

A deep neural network (DNN) (with an arbitrary two hidden layers node structure—DNN[a] in table 2) was initially applied to numerical data alone. In comparison with the linear classifier, test AUC-ROC and recall (for defaulted loans) scores improved to 68% and 67%, respectively. This indeed shows how more advanced feature combinations improve the predictive capabilities of the model. The improvement was expected, as the complexity of the phenomenon described by the target label surely implies more elaborated features and feature combinations than those originally provided to the model.

The DNN was then refined with a grid search on node numbers $n_1$, $n_2$ for the two hidden layers, with results shown in figure 3. The grid search was run over all combinations of values from the sets $n_1 \in \{5, 10, 15, 20, 30\}$, $n_2 \in \{1, 3, 5, 10\}$ and by applying a high level of dropout regularization (20%). The level of dropout regularization was empirically chosen from a [0%, 30%] range, this is a reasonable range for this type of models often found in the literature [27]. The strong regularization aimed to reduce the DNN's intrinsic tendency to overfit, leading to a more robust and general model infrastructure. Results on the test set were indeed verified to be largely in line throughout the grid search, suggesting a model which is robust in the context of hyperparameter tuning.

Results for two network structures selected from the grid search (together with DNN[a]—arbitrary two hidden layers node structure) are described in table 2. These network structures are selected, as their results display the desirable properties of stable AUC-ROC and high recall on defaults.

## 3.2. Model explainability and interpretation

In the spirit of good practice in artificial intelligence and machine learning, we delve deeper into the best performing model for the second phase. DNNs can replicate more complex functions, but one often risks to overfit or overlook major flaws in the model's understanding of the data. On the other hand, by deploying methods for model interpretation one can understand which features the model considers and reason why on the basis of domain knowledge and statistics. We examine variable importance for the model on out-of-sample test data as per the method in ch. 17 of [28]. This consists of shuffling one feature at a time and monitoring the change in model loss with respect to the loss for the original data. We extended the method to look at the change in metrics such as AUC-ROC and recall, by modifying the measure to account for the different interpretation of AUC-ROC increase (low feature importance) versus loss

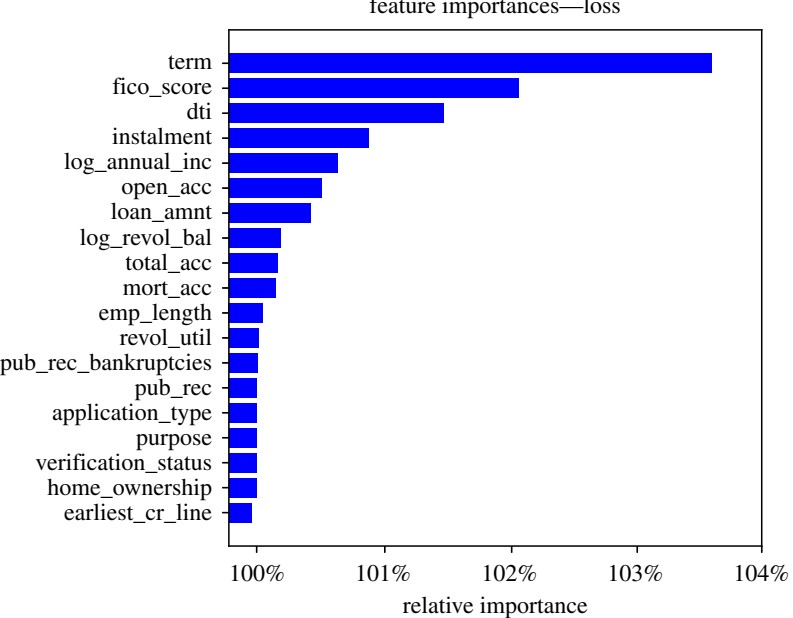

**Figure 4.** Feature importance from loss increase due to individual feature randomization.

increase (high feature importance—the randomization of the feature strongly affects the model's ability to predict). We then ranked the features by importance and represented their individual importances in figures 4 and 5. Before reasoning through the importance ranking of the features we note that the horizontal axis in figures 4 and 5 is presented in log scale, this is due to the high level of dropout the model was trained with. This causes the model to be quite solid against feature randomization, hence the scaling is necessary as the relative changes in loss value, although significant, do not deviate greatly from one (as this is a fraction of the reference loss/measure).

We note a strong overlap in the top features for the model, independently of the metric used. We now focus on the loss-based plot in figure 4 for a descriptive interpretation. We first note the 'term (of the loan)' feature as ranked first by all metrics, this is definitely expected as an increase in loan term implies higher interest rates, longer duration risk as well as a longer term exposure to the financial stability of the individual, which has more time to change and deteriorate from the state it was in when the loan was issued.

The 'FICO score' is ranked second. This was indeed expected to be a very meaningful feature as the FICO score is a widely used measure of an individual's creditworthiness. This combines many pieces of information through a finely tuned model. The large number of informational items it contains, together with the advanced modelling, explains why this feature was expected to be ranked among the top ones.

The third feature by rank in figure 4 is the debt to income ratio. This was also expected to be a relevant feature as it represents an individual's level of debt (this implies already pending repayments and usually lack of liquidity) as a fraction of his current income (this is the amount of cash flow that should allow the individual to repay his debts over time). This ratio also shows how much leverage one has in relation to his socio-economic status.

The loan amount is the fourth and last most relevant feature (we see a significant drop in importance following this). Clearly, not only the amount of debt in relation to income matters but also the amount of the current loan. The size of the loan influences the ability to repay it in case of distress and is an indicator of the lack (or need) of liquidity of the individual.

The 'purpose', 'verification_status', 'application_type' and 'home_ownership' features have no influence on the model as they are categorical features and have been set to be ignored by the model in the present work.

Features which are less intuitive in their relation to default are lowest in the rank, such as 'emp_length', 'earliest_cr_line', 'pub_rec' and 'total_acc'. This confirms that the model is interpreting the phenomenon in a sensible way in relation to domain knowledge and human reasoning.

We now show partial dependence profiles, as per ch. 18 of [28], for the four most relevant features in figure 4. These plots show the effect on the default probability of varying each feature across a range (here between its minimum and maximum in the test sample), given all other features stay the same. As the

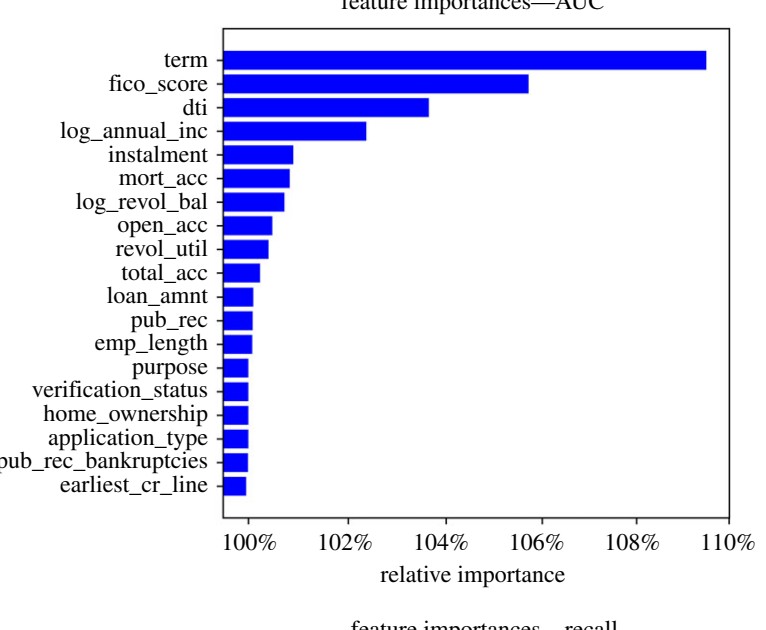

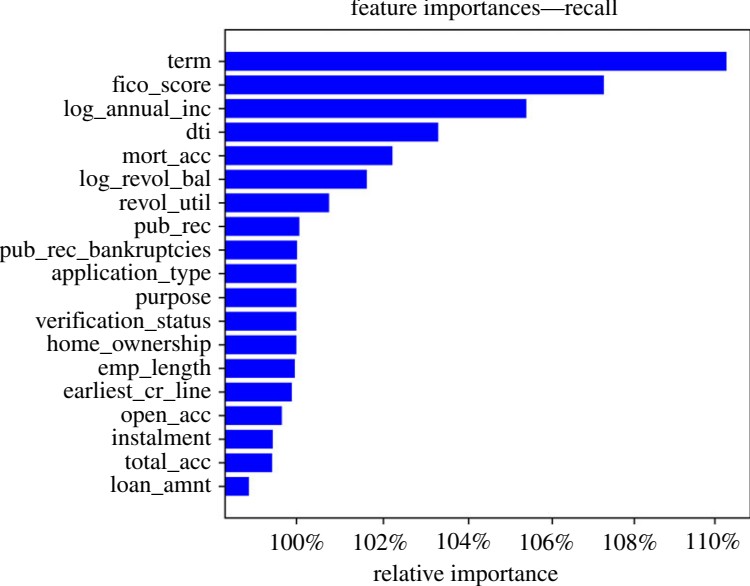

**Figure 5.** Feature importance from AUC and recall decrease from individual feature randomization.

'log_annual_inc' (log annual income of the individual) feature is at the top in both plots in figure 5, which are based on measures directly depend on probability, we investigate its partial dependence profile (PDP) plot as well in figures 6 and 7. We expect lower income to be associated with higher default probability.

We expect, especially for the 'term' and 'FICO score' features which are consistently ranked at the top in figures 4 and 5, to have a clear increase or decrease of the default probability as a function of the feature. Their plots are presented in figures 8 and 9. The 'loan_amnt (loan amount—of the current loan)' feature yields perhaps the least informative PDP plot, the feature is deemed less relevant in terms of AUC-ROC and recall. This can be associated with the feature's distribution being relevant for the loss function, but its probability being less informative for AUC-ROC and recall. PDP plots for the 'loan_amnt' and 'dti' features are presented in figures 10 and 11.

Figure 12 is a representation of one of the weight instances of the fully trained DNN network with node numbers $[n_1 = 5, n_2 = 3]$ from the grid search described in §2.2.2 over the range of node numbers described in §3.1.2.3. The network representation in figure 12 encodes the weight of each link in the fully connected layer as line thickness. Node size and colour are indicative of the normalized sum of outgoing weights from the node. This representation clearly constitutes an approximation, as the nodes contain nonlinearities, but it still provides a useful visual interpretation and stability check tool.

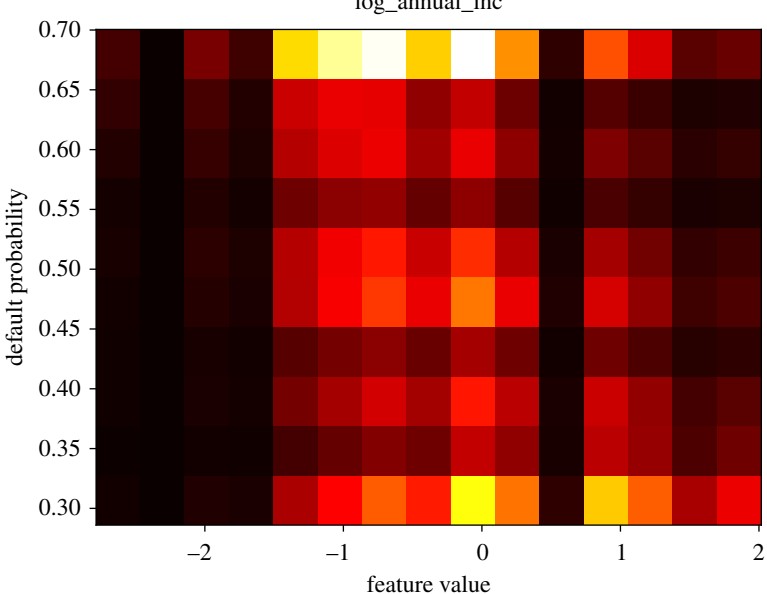

**Figure 6.** Partial dependence profile for the 'log annual income' feature.

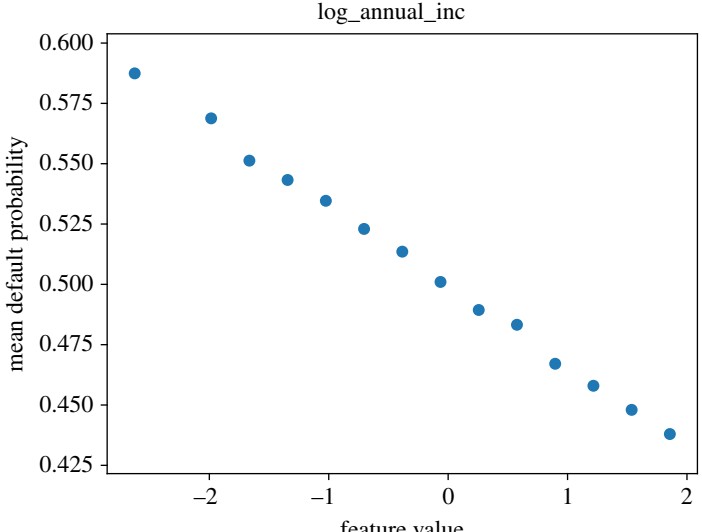

**Figure 7.** Partial dependence profile average for the 'log annual income' feature.

## 3.3. Two phases analysis for 'small business' category

The 'purpose' feature described in §2.2 provides information about the purpose for which the loan was requested. The small business class of this feature is of particular interest here. This loan category was observed to have the highest fraction of defaulted loans among all categories and the least likelihood to survive throughout the lending term period [12]. Furthermore, this purpose is arguably different from the others and is more business-focused, rather than just a personal loan.

We, therefore, decided to look at this category in isolation, although it was included in the entire dataset used for the analyses described in the previous sections.

### 3.3.1. First phase: small business training data only

LR and SVMs were trained and tested on 'small business' loans alone, with results summarized in table 3. Two grid searches were trained for LR; one maximizes AUC-ROC while the other maximizes recall macro. The former returns an optimal model with $\alpha = 0.1$, training AUC-ROC score $\simeq 88.9\%$ and

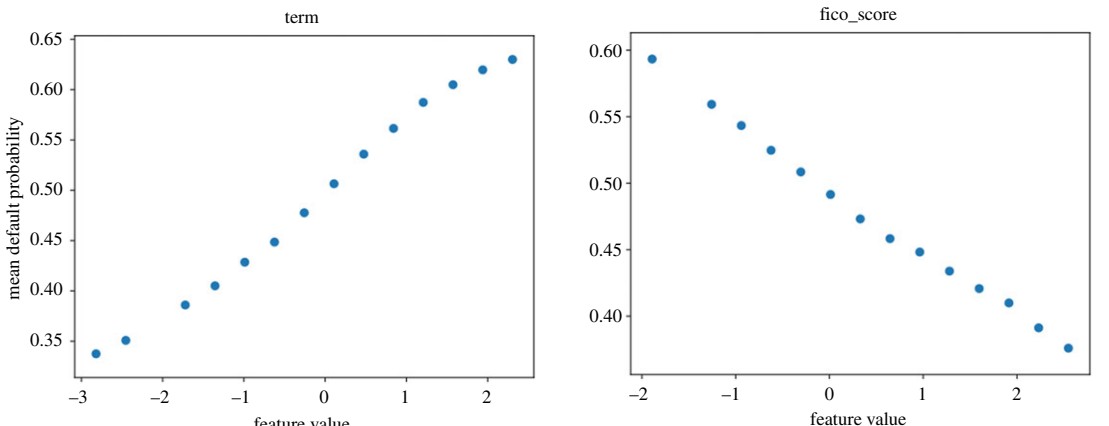

**Figure 8.** Partial dependence profiles for the 'term' and 'FICO score' features.

**Figure 9.** Partial dependence profile averages for the 'term' and 'FICO score' features.

test AUC-ROC score $\simeq 65.7\%$. Individual recall scores are $\simeq 48.0\%$ for rejected loans and 62.9% for accepted loans. The discrepancy between the training and test AUC-ROC scores indicates overfitting to the data or the inability of the model to generalize to new data for this subset. The latter grid search returns results which somewhat resemble the former one. Training recall macro is $\simeq 78.5\%$ while test recall macro is $\simeq 52.8\%$. AUC-ROC test score is 65.5% and individual test recall scores are 48.6% for rejected loans and 57.0% for accepted loans. This grid's results again show overfitting and the inability of the model to generalize. Both grids show a counterintuitively higher recall score for the underrepresented class in the dataset (accepted loans) while rejected loans are predicted with recall lower than 50%, worse than random guessing. This might simply suggest that the model is unable to predict for this dataset or that the dataset does not present a clear enough pattern or signal.

SVMs perform poorly on the dataset in a similar fashion to LR. Two grid optimizations are performed here too, in order to maximize AUC-ROC and recall macro, respectively. The former returns a test AUC-ROC score of 89.3% and individual recall scores of 47.8% for rejected loans and 62.9% for accepted loans. The latter grid returns a test AUC-ROC score of 83.6% with individual recall scores of 46.4% for rejected loans and 76.1% for accepted loans (this grid actually selected an optimal model with weak L1 regularization). A final model was fitted, where the regularization type (L2 regularization) was fixed by the user and the range of the regularization parameter was shifted to lower values in order to reduce underfitting of the model. The grid was set to maximize recall macro. This yielded an almost unaltered AUC-ROC test value of $\simeq 82.2\%$ and individual recall values of 47.3% for rejected loans and 70.9% for accepted loans. These are slightly more balanced recall values. However, the model is still clearly unable to classify the data well, this suggests that other means of evaluation or features could have been used by the credit analysts to evaluate the loans. The hypothesis is reinforced by the discrepancy of results with those described in §3.2 for the whole dataset. It should be noted, though, that the data for small business loans includes a much lower number of samples than that described in §3.1.1, with less than $3 \times 10^5$ loans and just $\approx 10^4$ accepted loans.

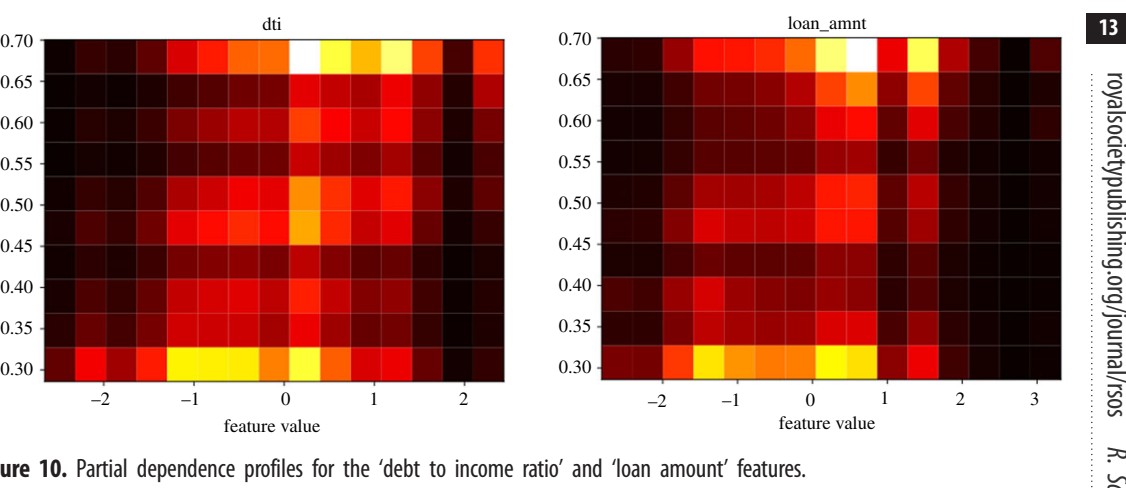

**Figure 10.** Partial dependence profiles for the 'debt to income ratio' and 'loan amount' features.

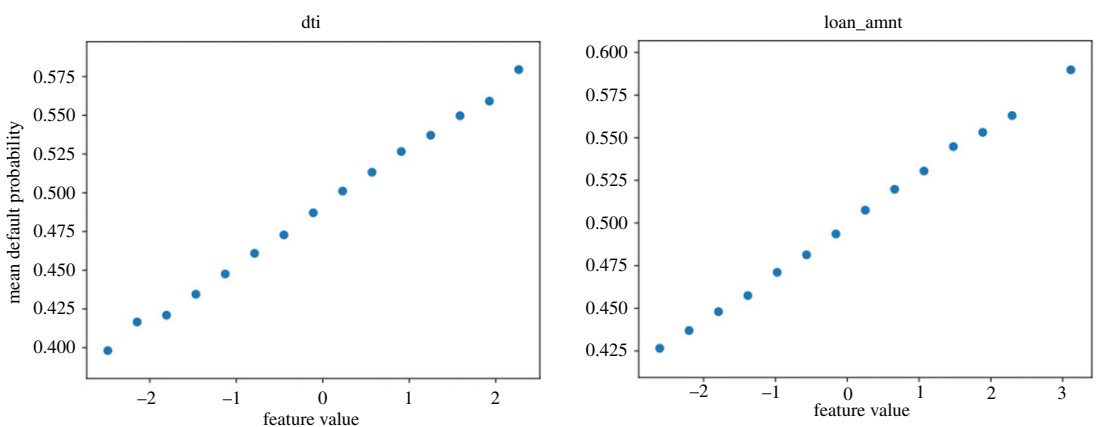

**Figure 11.** Partial dependence profile averages for the 'debt to income ratio' and 'loan amount' features.

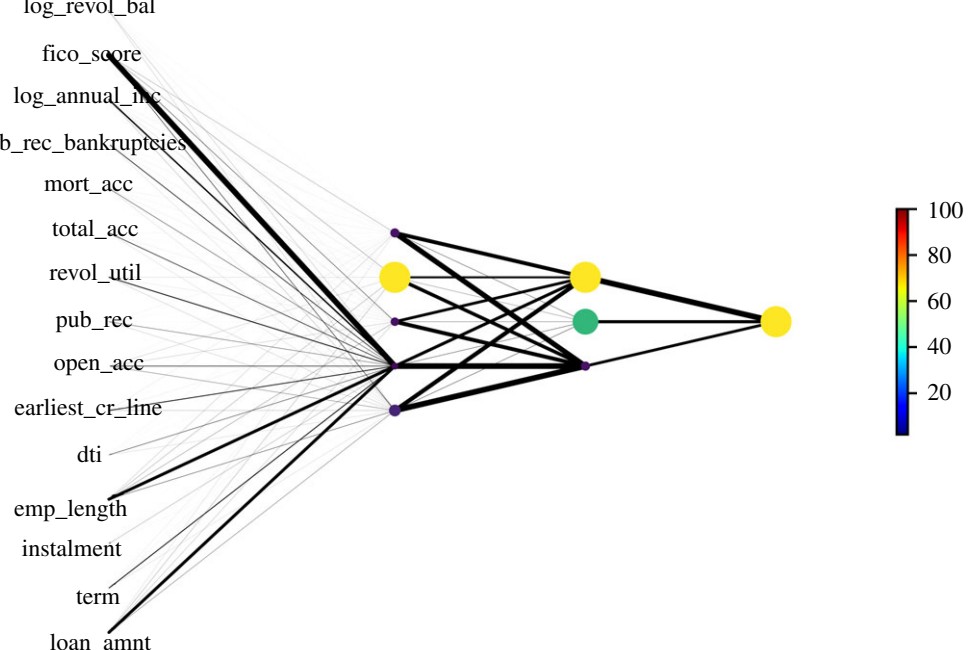

**Figure 12.** Neural network representation with node size and colour representing total outgoing weight and edge width proportional to the weight. The DNN represented is with node numbers $[n_1 = 5, n_2 = 3]$ and *tanh* nonlinearities.

**Table 3.** Small business loan acceptance results and parameters for SVM and LR grids trained and tested on the data's 'small business' subset.

| model | grid metric | $\alpha$ | training score | AUC test | recall rejected | recall accepted |
|-------|-------------|----------|----------------|----------|-----------------|-----------------|
| LR | AUC | 0.1 | 88.9% | 65.7% | 48.5% | 62.9% |
| LR | recall macro | 0.1 | 78.5% | 65.5% | 48.6% | 57.0% |
| SVM | recall macro | 0.01 | — | 89.3% | 47.8% | 62.9% |
| SVM | AUC | 10 | — | 83.6% | 46.4% | 76.1% |

**Table 4.** Small business loan acceptance results and parameters for SVM and LR grids trained on the entire dataset and tested on its 'small business' subset.

| model | grid metric | $\alpha$ | training score | AUC test | recall rejected | recall accepted |
|-------|-------------|----------|----------------|----------|-----------------|-----------------|
| LR | AUC | 1 | 89.0% | 71.9% | 53.5% | 60.2% |
| LR | recall macro | 0.1 | 77.9% | 71.7% | 54.0% | 59.9% |
| LR | fixed | 0.001 | 80.0% | 71.1% | 55.2% | 65.2% |
| LR | fixed | 0.0001 | 80.1% | 71.0% | 55.9% | 62.9% |
| SVM | recall macro | 0.01 | — | 77.5% | 52.6% | 68.4% |
| SVM | AUC | 10 | — | 89.0% | 97.3% | 43.3% |

### 3.3.2. First phase: all training data

Given the poor performance of the models trained on the small business dataset and in order to leverage the large amount of data in the main dataset and its potential to generalize to new data and to subsets of its data, LR and SVMs were trained on the whole dataset and tested on a subset of the small business dataset (the most recent loans, as by the methodology described in §2.2). This analysis yields significantly better results, when compared to those discussed in §3.3.1. Results are presented in table 4.

The results presented in table 4 for LR still present consistently higher recall for accepted loans. There is an apparent credit analyst decision bias towards rejecting small business loans. This could, though, be explained as small business loans have a higher likelihood of default, hence they are considered more risky and the model, trained on all the data, does not have this information. Information on loan defaults is present as a label only in default analysis, as no data are present for rejected loans. Future works might input the percentage of defaulted loans corresponding to the loan purpose as a new feature and verify whether this improves the model.

Results for SVMs are in line with those for LR. The grid trained to maximize AUC-ROC is clearly overfitting the rejected class to maximize AUC-ROC and should be discarded. Results for the grid maximizing recall macro follow the same trend of those from LR. Recall scores are slightly more unbalanced. This confirms the better performance of LR for the prediction task, as discussed in §3.1.1.

### 3.3.3. Second phase

LR and SVMs were trained on accepted loan data in order to predict defaults of loans with 'small business' purpose. Analogously to the analysis discussed in §3.3.1, the models were trained and tested on small business data alone. Results for models trained on small business data alone are presented in table 5. Results for LR are slightly worse and more unbalanced in individual recall scores than those presented in §3.1.2; this can be explained by the smaller training dataset (although more specific, hence with less noise). Surprisingly, again, the underrepresented class of defaulted loans is better predicted. This could be due to the significant decay of loan survival with time for small business loans; these data are obviously not provided to the model, hence the model might classify as defaulting, loans which might have defaulted with a longer term. Alternatively, most defaulting loans could be at high risk, while not all risky loans necessarily default, hence giving the score imbalance. Maximizing AUC-ROC in the grid search yields best and most balanced results for LR in this case. Analogously to the analysis in §3.3.1, class imbalance is strong here; defaulted loans are ≈3% of the

**Table 5.** Small business loan default results and parameters for SVM and LR grids trained and tested on the data's 'small business' subset.

| model | grid metric | $\alpha$ | training score | AUC test | recall defaulted | recall paid |
|-------|-------------|----------|----------------|----------|------------------|-------------|
| LR | AUC | 0.1 | 64.8% | 66.4% | 65.2% | 57.4% |
| LR | recall macro | 0.01 | 60.4% | 65.3% | 64.6% | 53.3% |
| SVM | recall macro | 0.01 | — | 59.9% | 59.8% | 58.8% |
| SVM | AUC | 0.1 | — | 64.2% | 50.8% | 65.8% |

**Table 6.** Small business loan default results and parameters for SVM and LR grids trained on the entire dataset and tested on its 'small business' subset.

| model | grid metric | $\alpha$ | training score | AUC test | recall defaulted | recall paid |
|-------|-------------|----------|----------------|----------|------------------|-------------|
| LR | AUC | 0.001 (L1) | 69.8% | 68.9% | 81.0% | 43.3% |
| LR | AUC | 0.001 | 69.7% | 69.2% | 86.4% | 35.0% |
| LR | recall macro | 0.001 | 64.2% | 69.2% | 86.4% | 35.0% |
| SVM | recall macro | 0.001 | — | 64.1% | 77.7% | 48.3% |
| SVM | AUC | 0.001 | — | 69.7% | 77.7% | 48.3% |

dataset. The better predictive capability on the underrepresented class might be due to loan survival with time and should be investigated in further works. Three threshold bands might improve results, where stronger predictions only are evaluated.

SVMs provide more balanced results, although worse overall, for this task. In both SVMs and LR we observe how stronger regularization, corresponding to higher values of $\alpha$, improves recall results on the test set for the overrepresented class. AUC-ROC test scores improve as well, suggesting an improvement in the model's ability to generalize.

Analogously to the analysis presented in §3.3.2, LR and SVMs were also trained on all the data and tested on small business data only, in order to leverage the larger datasets, which might share signals with its 'small business' subset. Results in this case, differ from those in §3.3.2, where an improvement was observed. Results are presented in table 6. The model poorly predicts fully paid loans, with a recall score even below 50%. This might suggest that the way these loans are screened is similar to that of other categories, but their intrinsic default risk is very different indeed. This is also observed in the discrepancy in loan survival between these loans and all other loan categories. Serrano-Cinca *et al.* [12]. The optimal parameters returned by the grid suggest weaker regularization than that for results in table 5. For predicting a subset of its data, stronger regularization might improve results; this could be verified in future works. It should be considered, though, that regularization might reduce the importance of a small subset of the data, such as that of small business loans. The fraction of the small business subset with respect to the complete dataset is roughly the same for loan acceptance ($\simeq$1.3%) and loan default prediction ($\simeq$1.25%). This indeed suggests a difference in the underlying risk of the loan and its factors.

As the conclusions about model generalization described in §4 can be drawn already by comparing LR and SVM models, DNNs are not considered for the small business dataset analysis in §3.3. DNNs are considered only for the purpose of improving model performance through more complex models and feature combinations, which is the theme of §3.1.

## 4. Conclusion

In this paper, we demonstrate that P2P loan acceptance and default can be predicted in an automated way with results above $\simeq$85% (rejection recall) for loan acceptance and above $\simeq$75% (default recall) for loan default. Given that the present loan screening has a resulting fraction of default around 20% (figure 1) we can infer that potentially the methodology presented in this paper could reduce the defaulting loans to 10% with positive consequences for the efficiency of this market. The best

performing tools were LR for loan acceptance and DNNs for loan default. The high recall obtained with linear models on replicating traditional loan screening suggests that there is significant room for improvement in this phase as well.

The loan grade and interest rate features were found to be the most relevant for predicting loan default in [12]. The current model tries to predict default without biased data from credit analysts' grade and assigned interest rate, hence these features are excluded. The DNN and LR models provide substantial improvements on traditional credit screening. A recall score significantly and robustly above 70%, with AUC-ROC scores $\simeq$70% for the DNN, improves even on the LR in [12]. The features provided to the model in our study generalize to any lending activity and institution, beyond P2P lending. The present work could, therefore, be augmented in order to predict loan default risk without the need for human credit screening.

The two phases model for all loan purposes described in §3.1 showed better performance overall, with well-balanced individual test recall scores for the second phase of 75% for defaulted loans. This shows the ability to predict well above 50% of defaults on loans screened and accepted by credit analysts, while not penalizing excessively the acceptance of well-performing loans. Training on the whole dataset for the first phase resulted in higher scores when applied to small business loans than when trained on small business loans alone. The opposite was true for the second phase, where default prediction was significantly better overall when trained on small business loans alone. This suggests a discrepancy between how credit analysts treat these loans and how they might be treated more efficiently, in terms of their default risk and characteristics. Neural networks were shown to significantly outperform the other models, suggesting that they might be used for default prediction, further to credit analyst screening. Neural networks could also be combined with LR in a conservative model, in order to mitigate their complex and not well-predictable nature. This and further data preprocessing and augmentation should be the subject of further work. We shall further extend our work to areas such as micro-financing in developing countries and loan-by-loan evaluation of loan portfolios for investment as well as to traditional lending. The integration of the present model with predictive modelling based on information filtering network techniques [29–32] will also be the subject of future research.

Data accessibility. Data and relevant code for this research work are stored in GitHub: https://github.com/jeremyDT/P2P-lending-with-AI and have been archived within the Zenodo repository: https://doi.org/10.5281/zenodo.3829483 [33] and the Dryad repository: https://doi.org/10.5061/dryad.qbzkh18cq [34].
Authors' contributions. J.D.T. conceived and designed the project and its implementation, acquired the data, carried out analysis and drafted the article. T.A. revised the article critically for important intellectual content, supervised and directed the research as well as the drafting of the article. All authors gave final approval for publication.
Competing interests. We declare we have no competing interests.
Funding. The authors acknowledge the EC Horizon 2020 FIN-Techproject (EC H2020-ICT-2018-2 825215) for partial support and useful opportunities for discussion. J.D.T. acknowledges support from EPSRC(EP/L015129/1). J.D.T. acknowledges Dr Guido Germano for useful feedback and discussions. T.A. acknowledges support from ESRC(ES/K002309/1), EPSRC(EP/P031730/1).

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
