## [Reviewer comments · Royal Society Open Science]

Review History

RSOS-191649.R0 (Original submission)

Review form: Reviewer 1

Is the manuscript scientifically sound in its present form?

Yes

Are the interpretations and conclusions justified by the results?

No

Is the language acceptable?

Yes

Do you have any ethical concerns with this paper?

No

Have you any concerns about statistical analyses in this paper?

Yes

Recommendation?

Major revision is needed (please make suggestions in comments)

Comments to the Author(s)

Deeper validation of obtained models is needed. Now we have lots of software for such validation (all these XAI tools) so it is really easy to use (short list below).

Without such validation we just promote the use of black-boxes like DNN without caring what is happening inside them.

Cathy O'Neil in her book Weapons of Math Destructions shows plenty of examples where this leads.

Here are three points that shall be addressed to do at least the based model validation.

1.

Authors do a lot of manual tuning for obtained models. There is not enough information what was the evaluation scheme.

I would expect that one evaluation of the data will be the out-of-time sample, external sample never saw by the model not never seen in the hyperparameter tuning.

2.

It is not clear if there was a nested CV performed or not. The grid search needs its own validation data, external from the out-of-sample validation data.

The validation shall be described in larger details.

3.

Reporting AUC or Recall is far not enough. The literature is full of examples of overtrained black-boxes that are not validated.

Authors shall do deeper post-hoc analysis of trained models with tools like Partial Dependency Profiles (https://pbiecek.github.io/PM_VEE/partialDependenceProfiles.html), Permutational feature importance (https://pbiecek.github.io/PM_VEE/featureImportance.html).

Lots of packages for R and python to do this validation.

All proposed models (logistic regression, DNN and SVN) shall be xrayed with these tools, as it is very easy oto overfit and create an unfair model.

Review form: Reviewer 2

Is the manuscript scientifically sound in its present form?

No

Are the interpretations and conclusions justified by the results?

No

Is the language acceptable?

Yes

Do you have any ethical concerns with this paper?

No

Have you any concerns about statistical analyses in this paper?

No

Recommendation?

Major revision is needed (please make suggestions in comments)

Comments to the Author(s)

A list of acronyms would be most useful, with the constituent words appearing in brackets next to each acronym.

Owing to its central role in the performed research, P2P should be defined at the very beginning of the manuscript.

Artificial neural networks belong to the computational (rather than artificial) intelligence paradigm.

The “significant step forward to applying Big Data and Artificial Intelligence techniques to P2P” is not made clear.

The review of the literature is not always pertinent to the main theme of the submission.

It is not clear why the two datasets have been used concurrently rather than independently. It is also not obvious how or why the results obtained can be transferred to other datasets.

Carrying on with the previous point: How uniform are the datasets over the kind of loan requested? Their discrepancies over parameters as well decision type should be clearly stated. Would it be to advantage to treat each category independently (in a “one against the others” fashion)?

Since imputation is below 10%, both the full and the reduced datasets could be used for training and testing, with the results compared and important derivations made.

Along the same lines, how uniform is the data over the kind of loan requested over the two datasets? Would it be to advantage to treat (consider, in terms of training/testing and results) each dataset independently, thus simplifying the problem as well as the implemented methods?

At present Section 2.b is quite descriptive, with the implemented procedure not being adequately detailed to allow its duplication by the interested reader.

Why have the specific ANN architectures been selected? Different neural network training criteria including

- pertinent architectures as well as nodes per layer;
- earlier termination of the training stage;
- cross-validation on the dataset (e.g. five-, 10- and/or leave-one-out cross-validation, with the folds being created either at random, or following ordering of the patterns with – for instance, for five-fold cross-validation - the 1st, 6th, 11th, ... pattern belonging to the first fold, the 2nd, 7th, 12th etc. to the second fold, and so forth to the fifth fold), so that each fold contains the same number of patterns which extend over the entire problem space, instead of just a “split between training and validation sets” should be investigated.

The authors should ensure that all methodologies, metrics, data handling techniques etc. are accompanied by the corresponding primary references.

Section 3 contains information that should be moved to the previous section.

If logistic regression is, indeed, as successful as stated in section 3.a(iii), there is no need for more complicated (especially non-parametric) methodologies, which can add redundant and distorting detail to the problem methodology/solution and are not directly/easily (or even at all) expressed in a direct/parametric fashion.

The authors should ensure that the dataset is stationary; in case this is not so, alternative methodologies and/or on-line (re)training should be implemented.

Class imbalance is not optimal for ANN training; the appropriate measures should be taken in order to avoid training (and, thus, also) testing bias.

For this kind of problems, on-line training would be advisable so as to ensure the appropriateness/capability of the ANN to handle the changing (in time) data characteristics.

There is a considerable distance between linear and deep neural networks. Why has not an alternative (in-between) ANN architecture also been tested?

In all cases, it should be ensured that the number of ANN free parameters (weights and biases) is smaller than the number of training patterns. The authors should check for overfitting in the DNNs used, especially as the datasets are small (in relation to the size of the ANN).

What distinguishes test from recall patterns? Has no validation set been used? Testing can, by no means, be implemented on data that has been used for training.

At times it is not clear whether the aim is to mimic human decision making or to optimise/maximise recall as well as prediction.

The choice of features for the first stage should be fully justified.

If properly constructed and trained, the ANNs should be able to accurately learn the training patterns, as well as predict the test data.

A list of acronyms would be most useful, with the constituent words appearing in brackets next to each acronym.

Owing to its central role in the performed research, P2P should be defined at the very beginning of the manuscript.

Artificial neural networks belong to the computational (rather than artificial) intelligence paradigm.

The “significant step forward to applying Big Data and Artificial Intelligence techniques to P2P” is not made clear.

The review of the literature is not always pertinent to the main theme of the submission.

It is not clear why the two datasets have been used concurrently rather than independently. It is also not obvious how or why the results obtained can be transferred to other datasets.

Carrying on with the previous point: How uniform are the datasets over the kind of loan requested? Their discrepancies over parameters as well decision type should be clearly stated. Would it be to advantage to treat each category independently (in a “one against the others” fashion)?

Since imputation is below 10%, both the full and the reduced datasets could be used for training and testing, with the results compared and important derivations made.

Along the same lines, how uniform is the data over the kind of loan requested over the two datasets? Would it be to advantage to treat (consider, in terms of training/testing and results) each dataset independently, thus simplifying the problem as well as the implemented methods?

At present Section 2.b is quite descriptive, with the implemented procedure not being adequately detailed to allow its duplication by the interested reader.

Why have the specific ANN architectures been selected? Different neural network training criteria including

- pertinent architectures as well as nodes per layer;
- earlier termination of the training stage;
- cross-validation on the dataset (e.g. five-, 10- and/or leave-one-out cross-validation, with the folds being created either at random, or following ordering of the patterns with – for instance, for five-fold cross-validation - the 1st , 6th, 11th, ... pattern belonging to the first fold, the 2nd, 7th, 12th etc. to the second fold, and so forth to the fifth fold), so that each fold contains the same number of patterns which extend over the entire problem space, instead of just a “split between training and validation sets” should be investigated.

The authors should ensure that all methodologies, metrics, data handling techniques etc. are accompanied by the corresponding primary references.

Section 3 contains information that should be moved to the previous section.

If logistic regression is, indeed, as successful as stated in section 3.a(iii), there is no need for more complicated (especially non-parametric) methodologies, which can add redundant and distorting detail to the problem methodology/solution and are not directly/easily (or even at all) expressed in a direct/parametric fashion.

The authors should ensure that the dataset is stationary; in case this is not so, alternative methodologies and/or on-line (re)training should be implemented.

Class imbalance is not optimal for ANN training; the appropriate measures should be taken in order to avoid training (and, thus, also) testing bias.

For this kind of problems, on-line training would be advisable so as to ensure the appropriateness/capability of the ANN to handle the changing (in time) data characteristics.

There is a considerable distance between linear and deep neural networks. Why has not an alternative (in-between) ANN architecture also been tested?

In all cases, it should be ensured that the number of ANN free parameters (weights and biases) is smaller than the number of training patterns. The authors should check for overfitting in the DNNs used, especially as the datasets are small (in relation to the size of the ANN).

What distinguishes test from recall patterns? Has no validation set been used? Testing can, by no means, be implemented on data that has been used for training.

At times it is not clear whether the aim is to mimic human decision making or to optimise/maximise recall as well as prediction.

The choice of features for the first stage should be fully justified.

If properly constructed and trained, the ANNs should be able to accurately learn the training patterns, as well as predict the test data.

Decision letter (RSOS-191649.R0)

20-Dec-2019

Dear Mr Turiel,

The editors assigned to your paper ("P2P Loan acceptance and default prediction with Artificial Intelligence") have now received comments from reviewers. We would like you to revise your paper in accordance with the referee and Associate Editor suggestions which can be found below (not including confidential reports to the Editor). Please note this decision does not guarantee eventual acceptance.

Please submit a copy of your revised paper before 12-Jan-2020. Please note that the revision deadline will expire at 00.00am on this date. If we do not hear from you within this time then it will be assumed that the paper has been withdrawn. In exceptional circumstances, extensions may be possible if agreed with the Editorial Office in advance. We do not allow multiple rounds of revision so we urge you to make every effort to fully address all of the comments at this stage. If deemed necessary by the Editors, your manuscript will be sent back to one or more of the original reviewers for assessment. If the original reviewers are not available, we may invite new reviewers.

- Data accessibility

<http://datadryad.org/submit?journalID=RSOS&manu=RSOS-191649>

- **Competing interests**

- **Authors' contributions**

- **Acknowledgements**

- **Funding statement**

Kind regards,

Royal Society Open Science Editorial Office
Royal Society Open Science
openscience@royalsociety.org

on behalf of Prof Marta Kwiatkowska (Subject Editor)
openscience@royalsociety.org

Associate Editor's comments:

The two reviewers have a number of suggestions and queries that should improve your manuscript - we would urge you to take their recommendations seriously, and be sure to not only include the requested updates, but provide an explanation of what changes have been made, or - even more importantly - if you choose not to include a change, explain why not. We'll look forward to receiving the revision in due course.

Comments to Author:

Reviewers' Comments to Author:

Reviewer: 1

Comments to the Author(s)

Deeper validation of obtained models is needed. Now we have lots of software for such validation (all these XAI tools) so it is really easy to use (short list below).

Without such validation we just promote the use of black-boxes like DNN without caring what is happening inside them.

Cathy O'Neil in her book *Weapons of Math Destruction* shows plenty of examples where this leads.

Here are three points that shall be addressed to do at least the based model validation.

1.

Authors do a lot of manual tuning for obtained models. There is not enough information what was the evaluation scheme.

I would expect that one evaluation of the data will be the out-of-time sample, external sample never saw by the model not never seen in the hyperparameter tuning.

2.

It is not clear if there was a nested CV performed or not. The grid search needs its own validation data, external from the out-of-sample validation data.

The validation shall be described in larger details.

3.

Reporting AUC or Recall is far not enough. The literature is full of examples of overtrained black-boxes that are not validated.

Authors shall do deeper post-hoc analysis of trained models with tools like Partial Dependency Profiles (https://pbiecek.github.io/PM_VEE/partialDependenceProfiles.html), Permutational feature importance (https://pbiecek.github.io/PM_VEE/featureImportance.html).

Lots of packages for R and python to do this validation.

All proposed models (logistic regression, DNN and SVN) shall be xrayed with these tools, as it is very easy to overfit and create an unfair model.

Reviewer: 2

Comments to the Author(s)

A list of acronyms would be most useful, with the constituent words appearing in brackets next to each acronym.

Owing to its central role in the performed research, P2P should be defined at the very beginning of the manuscript.

Artificial neural networks belong to the computational (rather than artificial) intelligence paradigm.

The "significant step forward to applying Big Data and Artificial Intelligence techniques to P2P" is not made clear.

The review of the literature is not always pertinent to the main theme of the submission.

It is not clear why the two datasets have been used concurrently rather than independently. It is also not obvious how or why the results obtained can be transferred to other datasets.

Carrying on with the previous point: How uniform are the datasets over the kind of loan requested? Their discrepancies over parameters as well decision type should be clearly stated. Would it be to advantage to treat each category independently (in a "one against the others" fashion)?

Since imputation is below 10%, both the full and the reduced datasets could be used for training and testing, with the results compared and important derivations made.

Along the same lines, how uniform is the data over the kind of loan requested over the two datasets? Would it be to advantage to treat (consider, in terms of training/testing and results) each dataset independently, thus simplifying the problem as well as the implemented methods?

At present Section 2.b is quite descriptive, with the implemented procedure not being adequately detailed to allow its duplication by the interested reader.

Why have the specific ANN architectures been selected? Different neural network training criteria including

- pertinent architectures as well as nodes per layer;
 - earlier termination of the training stage;
 - cross-validation on the dataset (e.g. five-, 10- and/or leave-one-out cross-validation, with the folds being created either at random, or following ordering of the patterns with - for instance, for five-fold cross-validation - the 1st, 6th, 11th, ... pattern belonging to the first fold, the 2nd, 7th, 12th etc. to the second fold, and so forth to the fifth fold), so that each fold contains the same number of patterns which extend over the entire problem space, instead of just a "split between training and validation sets"
- should be investigated.

The authors should ensure that all methodologies, metrics, data handling techniques etc. are accompanied by the corresponding primary references.

Section 3 contains information that should be moved to the previous section.

If logistic regression is, indeed, as successful as stated in section 3.a(iii), there is no need for more complicated (especially non-parametric) methodologies, which can add redundant and distorting detail to the problem methodology/solution and are not directly/easily (or even at all) expressed in a direct/parametric fashion.

The authors should ensure that the dataset is stationary; in case this is not so, alternative methodologies and/or on-line (re)training should be implemented.

Class imbalance is not optimal for ANN training; the appropriate measures should be taken in order to avoid training (and, thus, also) testing bias.

For this kind of problems, on-line training would be advisable so as to ensure the appropriateness/capability of the ANN to handle the changing (in time) data characteristics.

There is a considerable distance between linear and deep neural networks. Why has not an alternative (in-between) ANN architecture also been tested?

In all cases, it should be ensured that the number of ANN free parameters (weights and biases) is smaller than the number of training patterns. The authors should check for overfitting in the DNNs used, especially as the datasets are small (in relation to the size of the ANN).

What distinguishes test from recall patterns? Has no validation set been used? Testing can, by no means, be implemented on data that has been used for training.

At times it is not clear whether the aim is to mimic human decision making or to optimise/maximise recall as well as prediction.

The choice of features for the first stage should be fully justified.

If properly constructed and trained, the ANNs should be able to accurately learn the training patterns, as well as predict the test data.

A list of acronyms would be most useful, with the constituent words appearing in brackets next to each acronym.

Owing to its central role in the performed research, P2P should be defined at the very beginning of the manuscript.

Artificial neural networks belong to the computational (rather than artificial) intelligence paradigm.

The “significant step forward to applying Big Data and Artificial Intelligence techniques to P2P” is not made clear.

The review of the literature is not always pertinent to the main theme of the submission.

It is not clear why the two datasets have been used concurrently rather than independently. It is also not obvious how or why the results obtained can be transferred to other datasets.

Carrying on with the previous point: How uniform are the datasets over the kind of loan requested? Their discrepancies over parameters as well decision type should be clearly stated. Would it be to advantage to treat each category independently (in a “one against the others” fashion)?

Since imputation is below 10%, both the full and the reduced datasets could be used for training and testing, with the results compared and important derivations made.

Along the same lines, how uniform is the data over the kind of loan requested over the two datasets? Would it be to advantage to treat (consider, in terms of training/testing and results) each dataset independently, thus simplifying the problem as well as the implemented methods?

At present Section 2.b is quite descriptive, with the implemented procedure not being adequately detailed to allow its duplication by the interested reader.

Why have the specific ANN architectures been selected? Different neural network training criteria including

- pertinent architectures as well as nodes per layer;
- earlier termination of the training stage;
- cross-validation on the dataset (e.g. five-, 10- and/or leave-one-out cross-validation, with the folds being created either at random, or following ordering of the patterns with – for instance, for five-fold cross-validation - the 1st, 6th, 11th, ... pattern belonging to the first fold, the 2nd, 7th, 12th etc. to the second fold, and so forth to the fifth fold), so that each fold contains the same number of patterns which extend over the entire problem space, instead of just a “split between training and validation sets” should be investigated.

The authors should ensure that all methodologies, metrics, data handling techniques etc. are accompanied by the corresponding primary references.

Section 3 contains information that should be moved to the previous section.

If logistic regression is, indeed, as successful as stated in section 3.a(iii), there is no need for more complicated (especially non-parametric) methodologies, which can add redundant and distorting

detail to the problem methodology/solution and are not directly/easily (or even at all) expressed in a direct/parametric fashion.

The authors should ensure that the dataset is stationary; in case this is not so, alternative methodologies and/or on-line (re)training should be implemented.

Class imbalance is not optimal for ANN training; the appropriate measures should be taken in order to avoid training (and, thus, also) testing bias.

For this kind of problems, on-line training would be advisable so as to ensure the appropriateness/capability of the ANN to handle the changing (in time) data characteristics.

There is a considerable distance between linear and deep neural networks. Why has not an alternative (in-between) ANN architecture also been tested?

In all cases, it should be ensured that the number of ANN free parameters (weights and biases) is smaller than the number of training patterns. The authors should check for overfitting in the DNNs used, especially as the datasets are small (in relation to the size of the ANN).

What distinguishes test from recall patterns? Has no validation set been used? Testing can, by no means, be implemented on data that has been used for training.

At times it is not clear whether the aim is to mimic human decision making or to optimise/maximise recall as well as prediction.

The choice of features for the first stage should be fully justified.

If properly constructed and trained, the ANNs should be able to accurately learn the training patterns, as well as predict the test data.

Author's Response to Decision Letter for (RSOS-191649.R0)

See Appendix A.

RSOS-191649.R1 (Revision)

Review form: Reviewer 1

Is the manuscript scientifically sound in its present form?

Yes

Are the interpretations and conclusions justified by the results?

Yes

Is the language acceptable?

Yes

Do you have any ethical concerns with this paper?

No

Have you any concerns about statistical analyses in this paper?

No

Recommendation?

Accept with minor revision (please list in comments)

Comments to the Author(s)

Minor things:

- Instead of 'Partial Dependency' it should be 'Partial Dependence'.
- The OX axes in Figures 4 and 5 should be improved
- Figures 6 and 7 should be complemented with line cart that show how the conditional average probability depends on the values of the variables.
- Links to the code should be in Chapter 6 and not in references
- References are very incoherent, they need to be made more consistent. In particular, remove all links to the DOI.
- The small inscriptions in Figure 2 are unreadable. It is worth to enlarge them.

Review form: Reviewer 2

Is the manuscript scientifically sound in its present form?

Yes

Are the interpretations and conclusions justified by the results?

Yes

Is the language acceptable?

Yes

Do you have any ethical concerns with this paper?

No

Have you any concerns about statistical analyses in this paper?

No

Recommendation?

Accept with minor revision (please list in comments)

Comments to the Author(s)

The manuscript has been improved to a satisfactory degree.

Decision letter (RSOS-191649.R1)

Dear Mr Turiel:

On behalf of the Editors, I am pleased to inform you that your Manuscript RSOS-191649.R1

entitled "P2P Loan acceptance and default prediction with Artificial Intelligence" has been accepted for publication in Royal Society Open Science subject to minor revision in accordance with the referee suggestions. Please find the referees' comments at the end of this email.

The reviewers and Subject Editor have recommended publication, but also suggest some minor revisions to your manuscript. Therefore, I invite you to respond to the comments and revise your manuscript.

- Ethics statement

- Data accessibility

If you wish to submit your supporting data or code to Dryad (<http://datadryad.org/>), or modify your current submission to dryad, please use the following link:
<http://datadryad.org/submit?journalID=RSOS&manu=RSOS-191649.R1>

- Competing interests

- Authors' contributions

- Acknowledgements

- Funding statement

Because the schedule for publication is very tight, it is a condition of publication that you submit the revised version of your manuscript before 14-May-2020. Please note that the revision deadline will expire at 00.00am on this date. If you do not think you will be able to meet this date please let me know immediately.

on behalf of Prof Marta Kwiatkowska (Subject Editor)
openscience@royalsociety.org

Associate Editor Comments to Author:

It appears a couple of minor queries are left to address, but otherwise the paper is ready for acceptance. Please provide a final revision incorporating these remaining changes.

Reviewer comments to Author:

Reviewer: 2

Comments to the Author(s)

The manuscript has been improved to a satisfactory degree.

Reviewer: 1

Comments to the Author(s)

Minor things:

- Instead of 'Partial Dependency' it should be 'Partial Dependence'.
- The OX axes in Figures 4 and 5 should be improved
- Figures 6 and 7 should be complemented with line chart that show how the conditional average probability depends on the values of the variables.
- Links to the code should be in Chapter 6 and not in references
- References are very incoherent, they need to be made more consistent. In particular, remove all links to the DOI.
- The small inscriptions in Figure 2 are unreadable. It is worth to enlarge them.

Author's Response to Decision Letter for (RSOS-191649.R1)

See Appendix B.

Decision letter (RSOS-191649.R2)

Dear Mr Turiel,

It is a pleasure to accept your manuscript entitled "P2P Loan acceptance and default prediction with Artificial Intelligence" in its current form for publication in Royal Society Open Science.

on behalf of Prof Marta Kwiatkowska (Subject Editor)
openscience@royalsociety.org

Appendix A

We thank the reviewers for the careful scrutiny and useful comments which have greatly benefitted the quality of this work.

Reviewer 1

1. Authors do a lot of manual tuning for obtained models. There is not enough information what was the evaluation scheme.
I would expect that one evaluation of the data will be the out-of-time sample, external sample never saw by the model not never seen in the hyperparameter tuning.

As now clarified in Section 2.b.ii 5-fold cross validation is used for hyperparameter tuning, looking at mean AUC and Recall across folds to select the best architectures for which we then report out of sample results.

2. It is not clear if there was a nested CV performed or not. The grid search needs its own validation data, external from the out-of-sample validation data.
The validation shall be described in larger details.

Indeed cross validation was now performed with Figure 3 displaying a heatmap visualisation of these. As explained in the paper (Section 2.b.ii), cross validation was used in order to avoid shrinking the training or test sets.

3. Reporting AUC or Recall is far not enough. The literature is full of examples of overtrained black-boxes that are not validated.
Authors shall do deeper post-hoc analysis of trained models with tools like Partial Dependency Profiles (https://pbiecek.github.io/PM_VEE/partialDependenceProfiles.html), Permutational feature importance (https://pbiecek.github.io/PM_VEE/featureImportance.html).
Lots of packages for R and python to do this validation.

NOTE: the above link is no longer active – I found the new one which is referenced in the paper.

We have inserted an extended explainability part in Section 3.b. Looked at feature importance for loss, AUC and Recall (chapter 17). All makes sense and is thoroughly discussed there. We then look at partial dependency plots (chapter 18) for the most meaningful features, according to the feature importance technique from chapter 17. PDPs do not look as meaningful for all features – the model is trained with 20% dropout so relatively solid to individual feature changes.

Reviewer 2

A list of acronyms would be most useful, with the constituent words appearing in brackets next to each acronym.

Now done, where each acronym or abbreviation is used (used in parentheses next to the acronym the first time it is used).

Owing to its central role in the performed research, P2P should be defined at the very beginning of the manuscript.

Now defined more rigorously in the "Introduction" section.

Artificial neural networks belong to the computational (rather than artificial) intelligence paradigm.

This is now quite a topic of discussion. We have modified the sentence that caused this comment, it is otherwise acceptable in our opinion as it is common jargon in both industry and academia.

The "significant step forward to applying Big Data and Artificial Intelligence techniques to P2P" is not made clear.

This is now clarified in the introduction by specifying the aspects of the work which make it so.

The review of the literature is not always pertinent to the main theme of the submission.

We have added to/modified the literature review and references according to suggestions from the reviewers requesting more technical references for the methods used. Examples of this are references for AUC, Tanh activation function, and the different gradient descent methods in Section 2.b.ii.

It is not clear why the two datasets have been used concurrently rather than independently. It is also not obvious how or why the results obtained can be transferred to other datasets.

We have now explained the datasets again in clearer way at the beginning of section 2.b we have also added scheme to section 2.b with a small explanation of the models and processing pipeline.

GROUP OF REVIEWER COMMENTS (answered collectively below): Carrying on with the previous point: How uniform are the datasets over the kind of loan requested? Their discrepancies over parameters as well decision type should be clearly stated. Would it be to advantage to treat each category independently (in a "one against the others" fashion)?

Since imputation is below 10%, both the full and the reduced datasets could be used for training and testing, with the results compared and important derivations made.

Along the same lines, how uniform is the data over the kind of loan requested over the two datasets? Would it be to advantage to treat (consider, in terms of training/testing and results) each dataset independently, thus simplifying the problem as well as the implemented methods?

We believe these questions are related perhaps to the lack of clarity in explaining the data, we thank the reviewer for showing us this part was unclear. We have now made sure to clarify the understanding of how the datasets are used in Section 2.b.

At present Section 2.b is quite descriptive, with the implemented procedure not being adequately detailed to allow its duplication by the interested reader.

We have now cleaned this section and made it more schematic also with the addition of the scheme from Figure 2 which helps outline how the data is meant to be processed and looked at.

Why have the specific ANN architectures been selected? Different neural network training criteria including

- pertinent architectures as well as nodes per layer;
- earlier termination of the training stage;
- cross-validation on the dataset (e.g. five-, 10- and/or leave-one-out cross-validation, with the folds being created either at random, or following ordering of the patterns with – for instance, for five-fold cross-validation - the 1st, 6th, 11th, ... pattern belonging to the first fold, the 2nd, 7th, 12th etc. to the second fold, and so forth to the fifth fold), so that each fold contains the same number of patterns which extend over the entire problem space, instead of just a "split between training and validation sets" should be investigated.

We now explain the tuning of nodes per layer in Section 2.b.ii, as well as cross validation tuning and training.

The authors should ensure that all methodologies, metrics, data handling techniques etc. are accompanied by the corresponding primary references.

Primary references for optimisation, activation functions, metrics and more were added in Section 2.b.ii mainly.

Section 3 contains information that should be moved to the previous section.

Now done, moved to Section 2.

If logistic regression is, indeed, as successful as stated in section 3.a(iii), there is no need for more complicated (especially non-parametric) methodologies, which can add redundant and distorting detail to the problem methodology/solution and are not directly/easily (or even at all) expressed in a direct/parametric fashion.

This is connected to the lack of understanding/clarity about how the datasets are used. We have clarified this in Section 2.b (Methods) and across the paper. The more complex methods are used in the **second** phase of the model, whilst the simple one works well for the model trivial task of loan screening in the first phase of the model.

The authors should ensure that the dataset is stationary; in case this is not so, alternative methodologies and/or on-line (re)training should be implemented.

See two questions below.*

Class imbalance is not optimal for ANN training; the appropriate measures should be taken in order to avoid training (and, thus, also) testing bias.

As most balancing techniques are not optimal and there is no straightforward way to adjust with the chosen loss we downsample the data (i.e. the overrepresented non-default class). We have tried oversampling, but this caused overfitting to the repeated data points. We have now added this discussion to Section 2.b.ii. Future work may try this with bootstrapping to make the examples less similar.

*For this kind of problems, on-line training would be advisable so as to ensure the appropriateness/capability of the ANN to handle the changing (in time) data characteristics.

This may be again related to clarities in the dataset. We do not expect non-stationarities in this type of data. If in the default rates, this should not matter for a well-trained classifier. In terms of online learning and re-training we do not have enough data yet/a long enough period yet but as the number of loans is growing super-linearly (Figure 1) regular training should anyway give more importance to recent loans. We are hence unable to notice the difference yet. Due to imbalance (as many loans take time to default) we need a year gap between training data and the present where we would run the model.

There is a considerable distance between linear and deep neural networks. Why has not an alternative (in-between) ANN architecture also been tested?

Other architectures are currently being tested and will be the subject of future work. The current work already analyses four families of models and delving into more would drive attention away from the focus of the paper on a first Deep Learning application to P2P lending and its discussion.

In all cases, it should be ensured that the number of ANN free parameters (weights and biases) is smaller than the number of training patterns. The authors should check for overfitting in the DNNs used, especially as the datasets are small (in relation to the size of the ANN).

This is checked for in the explainability part Section 3.b where we open up the model and analyse features and more. As a side point, training parameters are in the order of 10^{2-3} and data is in the 10^{4-5} .

What distinguishes test from recall patterns? Has no validation set been used? Testing can, by no means, be implemented on data that has been used for training.

As explained above, we use cross validation for hyperparameter tuning and an out of sample test set to show results for the selected models. We have now outlined this more in detail in Section 2.b.ii and 3.a.v.

At times it is not clear whether the aim is to mimic human decision making or to optimise/maximise recall as well as prediction.

We have again clarified this by inserting the scheme in the Methods Section 2.b and better defined the difference between first and second phase. The first phase aims to mimic human decisions of acceptance/rejection as they are its target label. The second phase aims to predict default risk based on factual default data.

The choice of features for the first stage should be fully justified.

We wish to clarify that there was very little choice as no features are excluded, but geographic ones as they are categorical features and bare no intrinsic meaning. These should be used, encoded with related information, in further work.

If properly constructed and trained, the ANNs should be able to accurately learn the training patterns, as well as predict the test data.

This is indeed shown, as the model correctly interprets the features (explainability Section 3.b) and the selected models achieve high scores on out of sample test data in Section 3.a.v.

Appendix B

Response to minor revisions

We thank the reviewers for the previous and current comments which have greatly contributed to improving our work.

Reviewer comments to Author:

Reviewer: 2

Comments to the Author(s)

The manuscript has been improved to a satisfactory degree. **OK (no need for changes)**

Reviewer: 1

Comments to the Author(s)

Minor things:

- Instead of 'Partial Dependency' it should be 'Partial Dependence'. **This has been modified throughout the paper**
- The OX axes in Figures 4 and 5 should be improved **These have been improved (see figures)**
- Figures 6 and 7 should be complemented with line charts that show how the conditional average probability depends on the values of the variables. **These have been added in the new Figures 7,9,11**
- Links to the code should be in Chapter 6 and not in references **Done (see chapter 6)**
- References are very incoherent, they need to be made more consistent. In particular, remove all links to the DOI. **Links have been removed throughout references if not strictly necessary and reference format has been changed to improve readability.**
- The small inscriptions in Figure 2 are unreadable. It is worth to enlarge them. **These have been enlarged and are now visible.**